# The Future of Newborn Genomic Testing

**DOI:** 10.3390/children10071140

**Published:** 2023-06-30

**Authors:** John D. Lantos

**Affiliations:** Department of Pediatrics, Mt Sinai School of Medicine, New York, NY 10029, USA; john.lantos@gmail.com

**Keywords:** ethics, genomics, innovative therapy, neonatology, health policy

## Abstract

Genome sequencing (GS) provides exciting opportunities to rapidly identify a diagnosis in critically ill newborns and children with rare genetic conditions. Nevertheless, there are reasons to remain cautious about the use of GS. Studies to date have been mostly in highly selected populations of babies with unusual clinical presentations. GS leads to diagnoses in many such infants. More rarely, it leads to beneficial changes in management. Parents and physicians whose babies meet these criteria and for whom GS is performed both find these results useful. The concern is this: we do not know how useful such testing will be in the general population. We can speculate that a number of problems will arise as the use of GS expands. First, the percentage of cases in which a valid molecular diagnosis is made will likely go down. The number of ambiguous results or false positives will rise. Genetic counseling will become more complex and challenging. We do not know the relative cost-effectiveness of whole genome, whole exome, or targeted panels in different populations. We do not know the relative contribution of a molecular diagnosis to the decision to withdraw life support. We will have to carefully evaluate the use of such testing in order to understand whether it truly improves outcome and survival or reduces symptoms in babies who are tested. Each of these concerns will require careful study of both the technology and the ethical issues to allow us to harness the potential of these new technologies while avoiding foreseeable problems. Studies are underway to see how the tests are used in general populations. These studies should generate important information to guide clinicians and policymakers. As part of informed consent, doctors should explain to parents that genetic results are not always straightforward. Sometimes, they confirm a diagnosis that was already suspected. Sometimes, they rule out a possible diagnosis. Sometimes, the results are ambiguous and difficult to interpret. Anticipatory discussions should try to give parents a realistic understanding of the likely impact of a genetic diagnosis. Diagnostic genomic testing for newborns is a science that is still in its infancy. More research is essential in order to establish how to personalize this promising but sometimes problematic tool.

## 1. Introduction

It has been twenty years since the human genome was first sequenced. Since then, tools have been developed to rapidly sequence genomes and exomes. We can choose which genes to sequence. Genome sequencing (GS) may soon become the most expeditious way to screen infants for genetic diseases. However, experience with diagnostic genome sequencing suggests reasons for caution. As we learn about the complexities of interpreting GS results, we are uncovering new clinical and ethical issues associated with this powerful technology.

Past research on ethical, legal, and social issues (ELSIs) in genomics has allowed for careful study of this ethically complex technology. Early studies of ethical issues focused on the psychological sequelae of incidental findings. Bioethicists were especially concerned about testing that found variants of uncertain significance, worried that such results would be difficult for parents to understand and could lead to negative psychological sequelae [1]. Those fears turned out to be overblown. Studies showed that, with careful counseling and informed consent, most parents do not have negative psychological sequelae, even when test results are disturbing or ambiguous. Instead, parents report that genomic results are useful even if they are non-diagnostic, ambiguous, or if they diagnose a disease for which there is no effective treatment [2]. With those issues largely resolved, we now face a set of second-generation ethical issues associated with GS for critically ill newborns. Three questions seem key. First, who should be tested? Second, how accurate are the test results? And, third, does testing actually lead to better outcomes?

## 2. Who Should Be Tested?

Most babies who are admitted to the NICU are not suspected to have a genetic disease [3]. Instead, the most common diagnoses are prematurity, sepsis, or problems related to asphyxia during labor or at birth. Many other babies are admitted to NICUs with congenital anomalies. For these two groups, diagnosis and clinical care are relatively straightforward. A precise genomic diagnosis might help families make future reproductive choices, but it is unlikely to influence the clinical care of the babies.

Occasionally, babies have metabolic abnormalities or unusual neurological findings. For these babies, a diagnosis is often made using standard biochemical or metabolic tests. Post-natal GS is reserved for situations in which such testing does not lead to a clear diagnosis.

These epidemiological facts about neonatal care today are crucial to interpreting the results of NGS testing as it is used today and to inform decisions about how it should be used in the future.

So, who should be tested? As noted, current studies focus on the highly selected groups of babies who are symptomatic but for whom standard diagnostic testing has not led to a specific diagnosis. At the other extreme, Francis Collins, former Director of the NIH, suggested that every baby should have their genome sequenced at birth and the results used to guide a lifetime of health interventions [4].

GS, like all clinical testing, should only be performed when, in the doctor’s opinion, it is likely to yield valuable information that could help in the clinical management of the patient. With GS, the most likely scenario leading to clinical benefit is one in which such testing leads to the diagnosis of a disease that had not been suspected or confirmed. In such cases, GS may lead to beneficial treatment or preventive screenings. Estimates of how often GS leads to diagnoses or changes in management vary widely. There are at least three good reasons for the variation. 

First, the rate of positive results depends heavily on the choice of patients who are tested. Many studies report results from cohorts of babies who are tested because they have symptoms suggesting a genetic disease. Such babies are obviously more likely to have positive test results than babies who are asymptomatic. But, we often do not know precisely how decisions are made about which babies to test. Put another way, the inclusion criteria for studies of the efficacy of GS are not well-defined or standardized. 

Second, the rate of positive findings may also differ in different studies because there is variation in the ways that a “genomic” or “molecular” diagnosis can be defined. The phrase “molecular diagnosis” usually refers to a situation in which genome sequencing reveals a genomic variant that either has been reported to be associated with a patient’s symptoms or in which it is biologically plausible that such a relationship exists. In most cases, however, the population prevalence of such variants is unknown. Thus, we do not know how many people have the variant and have no symptoms. As a result, the nature of the causality is often an imperfect estimate, based on biological plausibility and prior reports of an association between the variant and the phenotype [5].

Finally, definitions of “change in management” also vary [6]. GS is often performed in infants with serious illness for whom changes in management are common, with or without molecular diagnoses. It may be difficult to determine whether a particular change in management was the direct result of information from the GS testing or whether the change would have occurred anyway. In addition, with GS, many proponents claim benefits that are not, strictly speaking, clinical benefits for the patient. For example, testing may end the diagnostic odyssey for parents. It may inform future reproductive decisions.

Because of variations in the populations of babies being studied, the criteria for claiming that a molecular diagnosis has been made, and the measurement of clinical utility in reported outcomes, it is difficult to know whether or how frequently GS will lead to real clinical benefit for either patients or their families. A study from The Netherlands that looked at every patient in the NICU or PICU in one hospital illustrates the complexities of estimating benefit [1]. Over 2 years, 1254 children <1 year of age were admitted to the NICU or PICU at one Dutch hospital. Of those, 24 (2%) were considered appropriate for GS because of a complex illness and diagnostic uncertainty; 7/24 had a finding on GS that was thought to explain the patients’ symptoms. The most frequent change in management that resulted from GS was to redirect care to comfort care. That occurred in 5/7 cases. There were no cases in which testing led to the provision of a new beneficial treatment.

Callaghan and colleagues conducted a meta-analysis of similar studies. They reported that there is no uniform definition of either a change in management or clinical utility [7]. The variation in the ways that outcomes are reported makes it difficult to compare studies. 

These ambiguities in the reporting of results make it difficult to know who to test. We know that the results of studies of GS in select populations of critically ill babies are unlikely to be generalizable to a less highly selected group of babies. We recognize the ambiguities of results in cases where it is used. These problems suggest the need for more research before we can say when and how GS should be used.

## 3. Does GS Improve Outcomes? 

The hope for NGS is that it will lead to unsuspected diagnoses of conditions for which treatments could improve important health outcomes. In most reported cohorts to date, that hope is rarely realized. Instead, when GS results lead to a change in management, that change is often either (1) a redirection of care towards palliation, (2) the referral of a patient to specialists for further evaluations (which may sometimes lead to benefit but the outcomes of such referrals are usually not described in papers), (3) a possible change in the parents’ future reproductive decisions (though the actual changes, too, are seldom described), or (4) the implementation of treatments that could, in many cases, have been implemented even without the genomic diagnosis. Each of these purported benefits raises ethical questions.

With regard to the redirection of care, the concern is whether the molecular diagnosis justifies a decision to provide comfort care instead of life-prolonging care. For example, one study reported that a molecular diagnosis of Noonan Syndrome led to a decision to initiate palliative care [8]. In another study, a diagnosis of Coffin–Siris syndrome resulted in the same [9]. These conditions would not ordinarily lead to a decision to initiate palliative car, as babies with the syndromes frequently have a good quality of life and are quite functional. The decisions to redirect care may well have been appropriate for those individual children but were likely based on multiple factors other than simply the genetic diagnosis. Furthermore, for many molecular diagnoses, there is a wide range of phenotypic findings, and children with precisely the same molecular genetic abnormality can have very variable outcomes. Even identical twins with the same molecular defect may have different phenotypes.

Most studies performed so far provide relatively little detail on how decisions about a redirection of care were reached. Clinical teams are typically thoughtful about the evaluation of genomic findings, disease severity, and family preferences before considering a withdrawal of support in the NICU and PICU setting. Still, it would be important for future studies to provide additional granularity on how molecular results might play a role in decisions to withdraw life support and to specify the relative contribution of the molecular diagnosis, as opposed to other clinical information, to the decision to withdraw life support.

The decision to withdraw life support after a molecular diagnosis also raises important questions about evaluations of cost-effectiveness. For example, the Baby Bear study analyzed the economics of GS and claimed that it was cost-saving. Much of the cost-saving was generated by 11 cases, in which life support was withdrawn after the return of GS results [10]. These decisions were judged to have saved expenditures on potentially futile care for babies with fatal illnesses. Similarly, in a multicenter Australian study, molecular diagnoses led to decisions to initiate palliative care in 14/55 (25%) patients with such diagnoses [11]. In both studies, details about the process of decision making were sparse. We do not know the weight given to the molecular diagnosis compared to that given to other factors. In some of the cases, the molecular diagnosis alone did not seem sufficient to justify withdrawal of treatment. Withdrawal of life support can certainly reduce healthcare costs, but the economic benefits need to be weighed against the ethical considerations that should always be primary in such decisions.

Many studies report cases in which a child receives a treatment after a molecular diagnosis, such as a study claiming that the initiation of treatment was a benefit of GS, but it seems that the child could and should have received the treatment based on clinical diagnosis and that the molecular diagnosis was either unnecessary or merely confirmatory rather than essential. Examples abound.

In a recent case series, a child with renal failure was found to have a genetic variant associated with hyperoxaluria [12]. In the past, that diagnosis would have been made by testing renal function and metabolites in serum and urine. Such traditional metabolic testing would likely have been quicker, more accurate, and more cost-effective than GS.

Another infant in the same report was found to have a variant associated with Dravet syndrome, a rare cause of epilepsy. Babies with Dravet need to be treated differently than patients with seizures caused by other diagnoses. Prior to GS, the diagnosis was made by testing different medications while monitoring the patient with a continuous EEG. With that method, doctors can observe the responses to different therapeutic agents. That approach to diagnosis can be carried out far more quickly than genomic sequencing. Furthermore, even with a molecular diagnosis showing a variant associated with Dravet, it would be necessary to perform continuous EEG testing and adjustments to treatment. The molecular diagnosis may not be precise. Genotype–phenotype correlations are not 100% [13].

Many studies report cases of children with non-specific developmental delays. The differential diagnosis in such cases is complex. There are many possible diagnoses. Doctors need to interpret individual tests with caution. Often, the diagnosis is only confirmed by a trial of therapy rather than a specific diagnostic test. GS results may add some certainty to the diagnosis, but they should rarely be the primary reason for changed management.

Many published series include similar case reports. It is difficult to know, from such reports, whether the molecular diagnosis was crucial, confirmatory, or relevant to the clinical decisions that were made for each patient [14]. As noted above, babies in the NICU have frequent changes in medication and management. Just because one occurs after a test result is obtained, it does not mean that the test result was a cause of the change. GS results are, sometimes, crucial. At other times, they are interesting but not essential. Sometimes, they can lead to unnecessary further testing or, as we explain below, even to the withdrawal of life support. Further careful research will be necessary to better clarify the appropriate role of such results in clinical decision making.

GS is a diagnostic modality. It should be compared with other diagnostic modalities to determine whether it is sensitive, specific, cost-effective, and clinically useful [15]. If so, it should certainly be used. But, because it is such a broad form of testing, and because many experts advocate its broad use, it is being used more indiscriminately than present results would warrant. In many cases, molecular diagnosis confirms a diagnosis for which there is no effective treatment. In such cases, GS could still be valuable. It might allow for more accurate genetic counseling so that families came make informed decisions about whether to have another baby or whether to avail themselves of prenatal or preimplantation genetic diagnosis.

One of the most difficult situations that arises from the use of GS is when the results are ambiguous [16]. In such cases, we have reason to suspect that a genomic variant is associated with the patient’s symptoms, but we just are not sure. The variant may be new. There may not be other cases with a similar genotype–phenotype association. Interpretation of GS results in such cases is as much an art as it is science. Ambiguities can be classified as false-positive tests or explained away with vague terms, such as “incomplete penetrance” or “variable expressivity” [17].

The complexity is perhaps best illustrated by cases in which patients with the same genotype have different phenotypes. This has been reported in many clinical situations [18]. It even occurs with identical twins. Clearly, there are things we do not understand about the ways that genes express themselves.

These difficulties in interpreting GS results are a reason why estimates of cost-effectiveness are complex. As noted, GS is not performed in isolation. It is hard to know when or whether it is the determining factor in clinical decisions. Efforts to show that it is cost-effective suffer from the same complexities as efforts to show the economic effect of any other single diagnostic test in the complex environment of neonatal intensive care [19].

No single test ever determines treatment. Doctors need to consider all the test results and then use their clinical judgment and expertise to decide which interventions are appropriate [20].

## 4. The Benefits (and Risks) of GS in Refining Prognostication

GS has benefits that go beyond direct clinical benefit. Sometimes, they help make prognostication more accurate. That, too, however, can be a double-edged sword. Prognostic information, like diagnostic information, is almost always probabilistic. Often, GS leads to a probabilistic estimate of the likelihood of a particular diagnosis, and that particular diagnosis may have a range of possible outcomes. For example, many genetic diseases are associated with a wide range of neurocognitive impairments. When we obtain the GS results, we may not know where the individual patient will ultimately fall along the wide spectrum of possible outcomes. Parents may not understand this and may make decisions about continuing or withdrawing life support based on their misunderstandings. 

GS often generates incidental findings, that is, genomic results that are not related to the primary clinical issues that were the motivation for ordering the test. For example, GS may reveal that patients have variants associated with adult-onset diseases. These can be thought of as prognostic tests, too, but prognostic tests not associated with the patients primary or current clinical problems. Incidental findings are common. In one study, 4.8% of pediatric patients who underwent NGS were found to have cancer susceptibility genes [21]. Future studies should focus on the ways that incidental findings impact patients and families.

## 5. How Often Does GS Harm Families?

There are many types of harms associated with GS. None are associated with the process of testing itself. That is risk free. The risks come from the ways in which test results are used.

Parents understandably may hope that GS will lead to a diagnosis and, even more, that the diagnosis will be of a curable condition. Most will be disappointed. It is hard to know whether that disappointment will have lasting psychological sequelae or how to measure that as a harm to be weighed against the benefits of a molecular diagnosis.

Some parents feel guilty when their child is diagnosed with an inheritable genetic disease [22]. Mothers and fathers may blame each other, leading to marital problems.

Sometimes, a false-positive or minimally predictive GS result leads parents to believe that their baby will die quickly. This can make emotional bonding difficult.

If doctors rely too heavily on GS, they may not perform other diagnostic tests, such as renal biopsies or continuous EEG monitoring. They may base their clinical assessments on the genomic tests rather than on clinical evaluation of the patient.

One potential harm is that GS results will be used to justify decisions to withdraw life support when the condition that is diagnosed would not justify such a decision. At the opposite extreme, some parents might wait to withdraw life support in cases where further treatment is futile or inappropriate because they want the false certainty of a genomic diagnosis.

Doctors need to help families understand that the most important factor in clinical decision making is the child’s medical condition. We do not need a genetic diagnosis in order to redirect care when a child is dying and further treatment will only prolong the dying process. In such cases, doctors must initiate difficult conversations about end-of-life options without imagining that more testing will make such conversations easier. After all, most GS results are non-diagnostic. Instead of confirming a diagnosis, they show that a patient does not have a suspected disease. As noted above, even in the most highly selected patient populations, GS leads to a molecular diagnosis in less than half of patients. If GS use becomes widespread and it is used in more general populations, the percentage of non-diagnostic tests will rise.

## 6. An Ethics Research Agenda for Genome Sequencing in Newborns

Research on ethical, legal, and social issues (ELSIs) in genomics has allowed for careful study of this ethically complex technology. Early studies examined the psychological sequelae of incidental findings, especially those of uncertain significance, that would lead to negative psychological sequelae [1]. They found that with careful counseling and informed consent, most parents do not have psychological sequelae, and they find genomic results useful, even if they are non-diagnostic or if there is no therapy for their child’s disease [2]. With those issues largely resolved, we now face a set of second-generation ethical issues associated with GS for critically ill newborns. We will need to develop and test strategies for testing in different populations. We will need to better define the benefits, costs, and harms of testing. We will need to differentiate the harms of the testing itself (which are negligible) from the harms that may accrue from misuse of the results. We will need to better understand the way doctors, parents, and patients use genomic information.

The likelihood of finding a genetic diagnosis will almost certainly be much lower when testing is performed in healthy babies or even in all NICU infants rather than in critically ill babies with a suspected genetic disease. Expansion of GS to healthy newborns is likely to generate results similar to those found in the pioneering BabySeq study [23]. In that study, 7.8% of healthy newborns had genomic variants that were classified as pathogenic. None of those babies were symptomatic. We will not know the predictive value of such testing until that cohort is followed for many years. In a critically ill newborn population, such high rates of incidental findings are often balanced by the finding of helpful diagnostic abnormalities. The frequency of potentially beneficial findings in a population of healthy newborns will be much lower, but these children will likely have a similar frequency of incidental abnormalities as well as more variants of unknown significance and even variants classified as clearly pathogenic despite being asymptomatic. Thus, future research in less highly selected populations will have to carefully study the unique counseling challenges that will arise when most results are ambiguous. Studies should also examine whether, in healthy populations, interrogation of more limited sets of genes will decrease the chance of unexpected ambiguous findings.

Two large studies, one in the UK and one in the US, may soon give answers to at least some of these questions. The Genomics England study plans to enroll 100,000 newborns over two years [24]. They will return results for 200 diseases that are thought to be clinically actionable. The Guardian Study in New York City also plans to screen 100,000 children and to test for and return results for 160 treatable diseases and 160 untreatable neurodevelopmental disorders [25]. These studies will also, almost certainly, suggest new questions to be investigated in future studies.

The definition of benefits from GS remains ambiguous. The best-case scenario is that testing leads to better treatments and improved outcomes. By this criterion, the jury is still out. No study to date of diagnostic GS in children has shown improved survival or shorter length of hospital stay in babies who received a diagnosis compared to those who did not.

Many studies choose different measures of benefit, including the above-mentioned surrogate endpoint of a change in management [26,27,28,29] or COM. Future research should add precision to the somewhat vague notion of what counts as a meaningful change in management (COM). COM is a complicated and ambiguous endpoint for three reasons. First, as Barrington pointed out, these are critically ill babies in ICUs for whom COMs occur all the time [14]. It is methodologically difficult to determine the precise cause of any COM. Second, COMs that do not lead to improved clinical outcome or survival may not be medically appropriate. They may just indicate false-positive tests or overuse of medical services. We know, for example, that false-positive results in newborn screening often increase parental stress without any benefit for the family or baby beyond ruling out the disease indicated by the false-positive test [30].

Future studies might clarify two aspects of the decision to initiate palliative care. First, they should examine whether a particular molecular diagnosis justifies the withholding of life-sustaining treatment. Second, they should examine whether decisions about withholding life-sustaining treatment are inappropriately delayed while awaiting genomic results in the belief that those results give certainty about prognosis. Often, prognostic estimates can be made accurately, based on the child’s clinical condition and hospital course.

## 7. Conclusions

GS provides exciting opportunities to rapidly identify a diagnosis in critically ill newborns and children with rare genetic conditions. But, there are good reasons to remain cautious about the broad implementation of these strategies in babies and children. At best, rapid GS leads to diagnoses in many infants in highly selected populations. It sometimes leads to beneficial changes in management. Parents and physicians both often find these results useful [31].

We do not know how useful such testing will be in the general population. It is almost inevitable that genetic counseling will be more challenging in a more general population. We do not know how often GS helps to improve outcomes and survival or reduce symptoms in babies who receive a molecular diagnosis. We do not know the relative cost-effectiveness of whole genome, whole exome, or targeted panels in different populations. We do not know the relative contribution of a molecular diagnosis to the decision to withdraw life support.

Each of these concerns will require careful study of both the technology and the ethical issues to allow us to harness the potential of these new technologies while avoiding foreseeable problems.

Studies are underway to see how the tests are used in general populations [24,25]. These studies should generate important information to guide parents, clinicians, and policymakers. The results will help doctors as they help parents reach a realistic understanding of the potential benefits and risks of GS testing.

## Data Availability

No original data in this study.

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
