# Peer review of "The Future of Newborn Genomic Testing"

_children, 2023, doi:10.3390/children10071140_

Round 1

Reviewer 1 Report

This paper makes a clinically informed critique of the rapid implementation of GS to neonatal populations that is nuanced and timely. The idea of a research agenda to interrogate “second generation ethical issues” is an especially strong and important distinction that I think is worth emphasizing earlier as the paper’s main contribution. The comments below are mostly aimed at increasing clarity and limiting redundancies.

-Many detailed and generalized factual statements lack citations (especially in the first half the paper). This makes it challenging to distinguish the author’s points from the information presented as the foundation of such arguments. Adding citations throughout would strengthen the paper and enhance clarity.

-Consider shortening the introduction by moving some details of limitations of existing research to later sections and specifically noting the introduction of a research agenda of second-generation ethical issues.

-Consider distinguishing the harm of the test from the harm of faulty expectations of the test (by clinicians and families) as this informs the last section on second-generation ethical issues.

-The concern raised about pending NGS delaying clinical decisions is especially important; consider reiterating in the final section about future research.

-The subsection header on how often NGS harm families might make more sense as “Assessing harms in NGS” to align with previous section.

-The point in the introduction about variability in molecular vs. genomic diagnoses is an important one and dovetails with the later point about incidental or secondary findings. In the ethics agenda section, consider adding a part on “What test” in addition to “Who to test” to help clarify questions to be addressed about the scope of GS, especially given variable reporting of secondary findings in existing studies and the degree of evidence needed to prognosticate in unselected populations when penetrance is largely unknown.

-The final section on withdrawal of life support also serves as a conclusion and includes information unrelated to that section header. Perhaps fold the withdrawal of life support info into preceding section and make this section a conclusion instead.

-The single reference to the analogous situation with current newborn screening platforms seems especially apt; consider if there are other places like the conclusion where this comparison might support the call for caution given extensive ethical critiques of how newborn screening has been expanded to include conditions without evidence of benefit.

-A point about discordance in twins is included twice and seems most relevant in the second reference.

Author Response

Thanks for these comments. They make the paper better. In response, I have

  1. Added many new citations
  2. shortened the introduction (and changed the organization of the paper)
  3. distinguished the harm of the test itself from harms that may accrue from the use of results
  4. emphasized the harm of delaying decisions while awaiting results
  5. changed all the section headings
  6. only talked about discordance in twins once
  7. I chose not to say more about traditional newborn screening. It is fascinating and important but beyond the scope of this paper.

Reviewer 2 Report

Thank you for the opportunity to review this manuscript. It makes some important practice points around the use of rapid genomic sequencing in NICU patients and the importance of clinical judgement and considering the overall clinical picture rather than developing an over-reliance on genomic tests.

The manuscript almost exclusively relates to diagnostic testing (rather than screening) in critically ill newborns (rather than healthy newborns). As there is a separate body of literature on the use of genomic sequencing as a screening tool in healthy newborns, I strongly recommend the title is revised to avoid misleading readers.

Overall, the manuscript can be improved with appropriately referencing key studies in the area as outlined below, but also those that have already explored parental perspectives, counselling challenges and ethical issues.

line 9 abstract: there appears to be a word missing after 'beneficial'

Lines 34-36: please provide some references for these statements

Line 46: the term 'mutation' is not recommended, please use 'variant'

Line 62: usually numbers at the start of sentences are written out as text

Line 71: there is also a move away from 'birth asphyxia' as a term

Line 78: this is an unreferenced statement, and it is not my clinical experience that many babies arrive with a diagnosis from prenatal testing 

Line 92: this references our own work I believe, though no reference is provided. Please note the infant was anuric and the diagnosis was completely unexpected and could not have been made in any other way, certainly not by collecting urine. There are many cases in the literature of this diagnosis being missed in similar scenarios with catastrophic outcomes following renal biopsy or transplant.

Line 98: this is an unsubstantiated statement. It is usual to give AEDs at least several days of trial until another AED is tried. There are many rapid testing programs that return results in much less time.

Line 102 onwards, not referenced, and speculation.

Line 116: I can only comment on apparent references to my own work but the child with congenital myasthenia had had many previous admissions to well regarded children's hospitals without the diagnosis being made clinically, emphasising the difficulty in rare disease diagnosis.

Line 140: there is a bracket missing

Line 144 onwards: none of the work is appropriately referenced.

Line 196: sentence split into two paragraphs

Line 217: full stop missing

Line 222: please define COM

Line 232 and 233: these are not referenced and not sure if they refer to patients from our cohorts but if so please note both infants had very significant other medical problems requiring major surgical interventions. The diagnosis of an underlying syndrome tipped the scale when deciding whether to proceed as you infer later: these are indeed complex and nuanced decisions.

Line 263 onwards: many of these concerns have been investigated and the relevant studies should be referenced.

Line 300: I am confused by the use of 'we', this is a single author paper?

Line 315: health should be healthy.

Line 318 onwards: please consider referencing the recent systematic review by Callahan et al PMID 35947384

Line 341, Section 7 reads as broad Conclusion which does not match the title of the section

See above for some minor issues.

Author Response

  1. I reviewed the paper and checked typos
  2. I have changed "mutation" to "variant." 
  3. The title is now "The Future of Newborn Genomic Testing." 
  4. The number of babies who arrive in the NICU with a genomic diagnosis is increasing
  5. The details about particular cases highlight the ways in which cases are reported in the literature without enough information for a reader to judge whether testing was necessary for diagnoses or changes in management.  
  6. The management of seizure is complicated. it is unclear whether molecular diagnosis will improve management.  In most centers, rapid return of results is not available. 
  7. General comment: my discussion of the way cases are reported is not meant to question the clinical decisions that were made.  It is only meant to highlight the ways in which cases are typically reported in case series of GS. 
  8. The paper has been reorganized so section headings are different. 

Round 2

Reviewer 2 Report

Thank you for the opportunity to review a revised version of this manuscript. Overall, the manuscript is much improved.

Some minor issues that should be corrected before publication:

Abstract: Sentence starting with 'More rarely..' the word 'and' seems superfluous. Para 2, sentence 2: US should be 'use'. Sentence 4: 'positive' should be 'positives'

Conclusion: penultimate sentence, 'beneficial and changes', delete 'and'

Author Response

I thank the reviewer and made all of the suggested changes.